# Sampling without Replacement Leads to Faster Rates in Finite-Sum Minimax Optimization

**Aniket Das** *
Indian Institute of Technology Kanpur
aniketd@iitk.ac.in

**Bernhard Schölkopf**
Max Planck Institute for Intelligent Systems
bs@tuebingen.mpg.de

**Michael Muehlebach**
Max Planck Institute for Intelligent Systems
michaelm@tuebingen.mpg.de

## Abstract

We analyze the convergence rates of stochastic gradient algorithms for smooth finite-sum minimax optimization and show that, for many such algorithms, sampling the data points *without replacement* leads to faster convergence compared to sampling with replacement. For the smooth and strongly convex-strongly concave setting, we consider gradient descent ascent and the proximal point method, and present a unified analysis of two popular without-replacement sampling strategies, namely *Random Reshuffling* (RR), which shuffles the data every epoch, and *Single Shuffling* or *Shuffle Once* (SO), which shuffles only at the beginning. We obtain tight convergence rates for RR and SO and demonstrate that these strategies lead to faster convergence than uniform sampling. Moving beyond convexity, we obtain similar results for smooth nonconvex-nonconcave objectives satisfying a two-sided Polyak-Łojasiewicz inequality. Finally, we demonstrate that our techniques are general enough to analyze the effect of *data-ordering attacks*, where an adversary manipulates the order in which data points are supplied to the optimizer. Our analysis also recovers tight rates for the *incremental gradient* method, where the data points are not shuffled at all.

## 1 Introduction

The approximate solution of large-scale optimization problems using first-order stochastic gradient methods constitutes one of the foundations of classical machine learning. However, emerging problems in machine learning go beyond pattern recognition and involve real-world decision making, where learning algorithms interact with unknown or even adversarial environments or are deployed in multi-agent settings. Decision making in such environments often involves solving a *minimax optimization* problem of the form $\min_{\mathbf{x}} \max_{\mathbf{y}} F(\mathbf{x}, \mathbf{y})$, whose analysis has been a focus of research in mathematics, economics, and theoretical computer science [38, 13, 10]. Recent examples of its applications in machine learning include adversarial learning [32, 52, 6], reinforcement learning [30, 58, 11, 42], imitation learning [14, 8, 24], and generative adversarial networks [18, 2]. In most of these applications, the objective $F(\mathbf{x}, \mathbf{y})$ has a finite-sum structure, i.e., $F(\mathbf{x}, \mathbf{y}) = 1/n \sum_{i=1}^{n} f_i(\mathbf{x}, \mathbf{y})$ where $n$ denotes the size of the dataset and each component function $f_i$ denotes the objective associated with the $i^{\text{th}}$ data point. The resultant problem is known as *finite-sum minimax optimization*:

---

*Currently at Google Research. Contact: ketd@google.com

36th Conference on Neural Information Processing Systems (NeurIPS 2022).

$$\min_{\mathbf{x} \in \mathbb{R}^{d_{\mathbf{x}}}} \max_{\mathbf{y} \in \mathbb{R}^{d_{\mathbf{y}}}} \frac{1}{n} \sum_{i=1}^{n} f_i(\mathbf{x}, \mathbf{y}). \tag{1}$$

When $n$ is large and the $f_i$'s are differentiable (which holds for most applications), approximate solutions to (1) are computed using stochastic gradient algorithms. These algorithms typically sample an index $i \in [n]$ at each iteration as per some specified sampling routine, and use the gradients of $f_i$ to compute the next iterate. Among these methods, perhaps the simplest and most commonly used algorithm is Stochastic Gradient Descent Ascent (SGDA), a natural extension of Stochastic Gradient Descent (SGD) for minimax optimization.

Similar to the stochastic optimization literature, analysis of stochastic minimax optimization often assumes that, at every iteration, the index $i$ is sampled *uniformly with replacement*. Analysis of the resulting algorithm closely parallels the analysis of SGD, and relies on the fact that uniform sampling leads to unbiased gradient estimates. Recent works [20, 31] have also extended this paradigm to i.i.d uniform sampling of mini-batches. While uniform sampling assumptions simplify theoretical analysis, practical implementations of these algorithms often deviate from this paradigm, and instead incorporate various heuristics, which are often empirically found to improve runtime. A common and notable heuristic is to replace uniform sampling by procedures that perform multiple passes over the entire dataset, and in each such pass (called an epoch), sample the data points *without replacement*. Thus, each data point is sampled exactly once in every epoch. These procedures are generally implemented using one of the following approaches:

**Random Reshuffling** (RR): Uniformly sample a random permutation of $[n]$ at the start of every epoch, and process the data points within that epoch as per the order specified by the permutation.

**Single Shuffling** or **Shuffle Once** (SO): Uniformly sample a random permutation at the beginning and reuse it across all epochs to order the data points.

**Incremental Gradient** (IG): Do not permute the data points at all and follow a fixed deterministic data ordering for every epoch.

Sampling without replacement is ubiquitous in both stochastic minimization [7, 48, 4] and stochastic minimax optimization [18, 2] as it often exhibits faster runtime than uniform sampling. However, these empirical benefits come at the cost of limited theoretical understanding, due to the absence of provably unbiased gradient estimates.

It is well known in the *optimization* literature that SGD with replacement has a tight rate of $O(1/nK)$ for smooth and strongly convex minimization [44, 25], where $n$ is the number of component functions and $K$ denotes the number of epochs. On the contrary, recent works on SGD without replacement for smooth and strongly convex minimization [1, 36, 34, 39] show that both RR and SO achieve a non-asymptotic rate of $\tilde{\mathcal{O}}(1/nK^2)$, once the number of epochs $K$ is larger than a certain threshold $K_0$ (usually polynomial in the condition number), and thereby converge faster than SGD with replacement. These rates match the lower bound of $\Omega(1/nK^2)$ for RR and SO established in prior works [43, 48], modulo logarithmic factors. For RR, prior works have also established a similar $\tilde{\mathcal{O}}(1/nK^2)$ rate for nonconvex objectives satisfying the Polyak-Łojasiewicz (PŁ) inequality [1, 34]. While the asymptotic behavior of IG has been known to the community for a long time in both smooth and non-smooth settings [5, 37], non-asymptotic $\tilde{\mathcal{O}}(1/K^2)$ convergence rates have been established quite recently [34, 39, 22], and are complemented by a matching $\Omega(1/K^2)$ lower bound [48].

## 1.1 Contributions

Although the empirical benefits of sampling without replacement have been substantiated for minimization, analysis of these methods for *minimax optimization* have received much less attention, despite being widely prevalent in many applications. Our work aims to fill this gap by analyzing these methods for minimax optimization. To this end, our main contributions are as follows:

**Unified analysis of RR and SO for smooth strongly convex-strongly concave problems:** We analyze RR and SO in conjunction with simultaneous *Gradient Descent Ascent* (GDA), calling the resulting algorithms GDA-RR and GDA-SO, respectively. Assuming the components $f_i$ are smooth and $F$ is strongly convex-strongly concave, we present a unified analysis of GDA-RR/SO and establish a convergence rate of $\tilde{O}(\exp(-K/5\kappa^2) + 1/nK^2)$ for both (where $\kappa$ is the condition

number). Comparing with lower bounds, we show that our rates are *nearly tight*, i.e., they differ from the lower bound only by an exponentially decaying term. Moreover, when $K \geq 10\kappa^2 \log(n^{1/2}K)$, the convergence rate matches the lower bounds for GDA-RR/SO, modulo logarithmic factors, and also converges provably faster than SGDA with replacement. Under the same setting, we obtain similar guarantees for the RR and SO variants of the *Proximal Point Method* (PPM), named PPM-RR and PPM-SO respectively. Our analysis for both GDA-RR/SO and PPM-RR/SO is general enough to cover smooth strongly monotone finite-sum variational inequalities, which covers minimization, minimax optimization, and multiplayer games.

**RR for smooth two-sided PŁ objectives:** We consider a class of nonconvex-nonconcave problems where the objective $F$ satisfies a *two-sided Polyak-Łojasiewicz* inequality. For such problems, we propose an algorithm that combines RR with two-timescale *Alternating Gradient Descent Ascent* (AGDA), which we call AGDA-RR. We show that AGDA-RR has a nearly tight convergence rate of $\tilde{O}(\exp(-K/365\kappa^3) + 1/nK^2)$ when the gradient variance is uniformly bounded. When $K \geq 730\kappa^3 \log(n^{1/2}K)$, this rate matches the lower bound (modulo logarithmic factors) and improves on the best known rates of with-replacement algorithms for this class of problems.

**Minimax optimization under data ordering attacks:** Our techniques for analyzing RR/SO generalize to the analysis of finite-sum minimax optimization under *data ordering attacks* [51]. These attacks target the inherent randomness assumptions of stochastic gradient algorithms, significantly increasing training time and reducing model quality, only by manipulating the order in which the algorithm receives data points, without performing any data contamination. To model these attacks, we propose the *Adversarial Shuffling* (AS) setup, where the data points are shuffled every epoch by a computationally unrestricted adversary. In this setup, we show that GDA and PPM (now called GDA-AS and PPM-AS) have a convergence rate of $\tilde{O}(\exp(-K/5\kappa^2) + 1/K^2)$ for smooth strongly convex-strongly concave objectives, and AGDA (now called AGDA-AS) has a rate of $\tilde{O}(\exp(-K/365\kappa^3) + 1/K^2)$ for two-sided PŁ objectives. We note that, compared to RR and SO, the convergence rate worsens by a factor of $1/n$ for large enough $K$. When $n$ is large (true for most applications), this slowdown significantly impacts convergence and thus, theoretically justifies the empirical observations in prior work [51]. We also establish that our analysis in the AS regime also applies to the Incremental Gradient (IG) variants of these algorithms (namely GDA-IG, PPM-IG, and AGDA-IG), and use this to show that our obtained rates for GDA-RR and AGDA-RR are nearly tight.

To the best of our knowledge, our work is the first to: 1) analyze RR, SO, and IG for strongly monotone unconstrained variational inequalities, 2) analyze RR and IG for a class of nonconvex-nonconcave minimax problems, 3) provably demonstrate the advantages of sampling without replacement for both these settings and justify its empirical benefits in a wide variety of problems ranging from minimization, minimax optimization to smooth multiplayer games, 4) analyze sampling without replacement under data-ordering attacks. Furthermore, unlike prior works on sampling without replacement for minimax optimization [55, 33], which are restricted to random reshuffling and require the component functions to be convex-concave, Lipschitz, and smooth, our analysis does not impose any restrictions on the components $f_i$ other than smoothness, allowing them to be arbitrary nonconvex-nonconcave functions.

## 2   Notation and Preliminaries

We work with Euclidean spaces $(\mathbb{R}^d, \langle ., . \rangle)$ equipped with the standard inner product $\langle \mathbf{x}_1, \mathbf{x}_2 \rangle$ and the induced norm $|\mathbf{x}|$. For any $\mathbf{x} \in \mathbb{R}^{d_\mathbf{x}}$ and $\mathbf{y} \in \mathbb{R}^{d_\mathbf{y}}$, we denote $\mathbf{z} = (\mathbf{x}, \mathbf{y}) \in \mathbb{R}^d$ where $d = d_\mathbf{x} + d_\mathbf{y}$. Moreover, for any $\mathbf{z}_1 = (\mathbf{x}_1, \mathbf{y}_1) \in \mathbb{R}^d$ and $\mathbf{z}_2 = (\mathbf{x}_2, \mathbf{y}_2) \in \mathbb{R}^d$, $\langle \mathbf{z}_1, \mathbf{z}_2 \rangle = \langle \mathbf{x}_1, \mathbf{x}_2 \rangle + \langle \mathbf{y}_1, \mathbf{y}_2 \rangle$ and $|\mathbf{z}_1|^2 = |\mathbf{x}_1|^2 + |\mathbf{y}_1|^2$. Whenever $\mathbf{z} = (\mathbf{x}, \mathbf{y})$ is clear from the context, we write $f(\mathbf{x}, \mathbf{y})$ as $f(\mathbf{z})$. We use $\mathbb{S}_n$ to denote the set of all permutations of $[n] = \{1, \ldots, n\}$. For any matrix $\mathbf{A}$, its operator norm is denoted by $|\mathbf{A}| = \sup_{|\mathbf{x}|=1} |\mathbf{A}\mathbf{x}|$. We use the $O$ notation to characterize the dependence of our convergence rates on $n$ and $K$, suppressing numerical and problem-specific constants such as $\kappa, \mu, \sigma$, etc. Additionally, we use the $\tilde{O}$ notation to suppress logarithmic factors of $n$ and $K$.

Our work studies finite-sum minimax optimization (1). Solutions to (1) are known as *global minimax points* of $F = 1/n \sum_{i=1}^n f_i$, which *we assume to always exist*. We also assume that the components $f_i$ are continuously differentiable, and hence, the same applies to $F$. This allows us to define the

*gradient operators* $\omega_i : \mathbb{R}^d \to \mathbb{R}^d$ *and* $\nu : \mathbb{R}^d \to \mathbb{R}^d$ *as follows:*

$$\omega_i(\mathbf{x}, \mathbf{y}) = [\nabla_{\mathbf{x}} f_i(\mathbf{x}, \mathbf{y}), -\nabla_{\mathbf{y}} f_i(\mathbf{x}, \mathbf{y})], \ \ \nu(\mathbf{x}, \mathbf{y}) = 1/n \sum_{i=1}^{n} \omega_i(\mathbf{x}, \mathbf{y}).$$

We also impose the following smoothness assumption on the components $f_i$.

**Assumption 1** (Component Smoothness). *The component functions $f_i$ are l-smooth, i.e., each gradient operator $\omega_i$ is l-Lipschitz*

$$|\omega_i(\mathbf{z}_2) - \omega_i(\mathbf{z}_1)| \le l\,|\mathbf{z}_2 - \mathbf{z}_1|.$$

*Consequently, the operator $\nu$ is also l-Lipschitz, i.e., $F$ is l-smooth.*

## 3 Analysis for Strongly Convex-Strongly Concave Objectives

In this section, we analyze two very popular without-replacement algorithms for finite-sum minimax optimization, Gradient Descent Ascent (GDA) without replacement and Proximal Point Method (PPM) without replacement. For each of these, we present a unified analysis of the Random Reshuffling (RR) and Shuffle Once (SO) variants (called GDA-RR/SO and PPM-RR/SO respectively). For a fixed $K > 0$, GDA-RR/SO approximately solves (1) by iterating over the entire dataset for $K$ epochs, and within each epoch, uses the operators $\omega_i$ to perform the following iterative update:

$$\mathbf{z}_i^k \leftarrow \mathbf{z}_{i-1}^k - \alpha \omega_{\tau_k(i)}(\mathbf{z}_{i-1}^k) \ \forall i \in [n], \tag{2}$$

where $\tau_k$ is a uniformly sampled random permutation of $[n]$ and $0 < \alpha < 1/l$ is a constant step-size. GDA-RR resamples $\tau_k$ at the start of every epoch, whereas GDA-SO samples it only once in the beginning. The details of both algorithms are presented in Algorithm 1. The Proximal Point Method without replacement is a closely related algorithm which, instead of performing gradient descent-style updates within an epoch, solves the following implicit update equation for $\mathbf{z}_i^k$:

$$\mathbf{z}_i^k = \mathbf{z}_{i-1}^k - \alpha \omega_{\tau_k(i)}(\mathbf{z}_i^k) \ \forall i \in [n]. \tag{3}$$

As before, $\tau_k$ is resampled at every epoch for PPM-RR, and sampled once and fixed for all epochs for PPM-SO. We present the details in Algorithm 2. The $l$-smoothness of $\omega_i$ along with the choice of $\alpha < 1/l$ ensures that $\mathbf{z}_i^k$ is uniquely defined, since it is a fixed point of the contraction mapping $\zeta(\mathbf{z}) = \mathbf{z}_{i-1}^k - \alpha \omega_{\tau_k(i)}(\mathbf{z})$. This method is actually a generalization of the (stochastic) proximal point method for minimization problems, and is popular for problems where (3) can be solved easily or in closed form. We refer the readers to Rockafellar [47], Patrascu and Necoara [41] for a review of this method and its connections to the original proximal point method for minimization.

### 3.1 Setting

We analyze GDA-RR/SO and PPM-RR/SO for smooth finite-sum strongly convex-strongly concave (or SC-SC) objectives. This allows us to formulate the minimax optimization problem for $F$ as a *root finding problem* for the gradient operator $\nu$, as described below.

**Assumption 2** (Strong Convexity-Strong Concavity). *The objective $F$ is $\mu$ strongly convex-strongly concave (or SC-SC), i.e., $F(., \mathbf{y})$ is $\mu$-strongly convex for any $\mathbf{y} \in \mathbb{R}^{d_{\mathbf{y}}}$ and $-F(\mathbf{x}, .)$ is $\mu$-strongly convex for any $\mathbf{x} \in \mathbb{R}^{d_{\mathbf{x}}}$.*

Assumption 2 has the following consequences for the gradient operator $\nu$:

**Lemma 1.** *Let $F$ satisfy Assumptions 1 and 2. Then, the gradient operator $\nu$ is $\mu$-strongly monotone:*

$$\langle \nu(\mathbf{z}_1) - \nu(\mathbf{z}_2), \mathbf{z}_1 - \mathbf{z}_2 \rangle \ge \mu\,|\mathbf{z}_1 - \mathbf{z}_2|^2 \ \forall\, \mathbf{z}_1, \mathbf{z}_2 \in \mathbb{R}^d.$$

*Furthermore, (1) admits a unique solution $\mathbf{z}^*$, which is also the unique solution of $\nu(\mathbf{z}^*) = 0$.*

Lemma 1 allows us to recast (1) for SC-SC objectives as the following *root finding problem*:

$$\text{Find } \mathbf{z} \in \mathbb{R}^d \text{ such that } \nu(\mathbf{z}) = 1/n \sum_{i=1}^{n} \omega_i(\mathbf{z}) = 0. \tag{4}$$

| **Algorithm 1:** GDA-RR/SO/AS | **Algorithm 2:** PPM-RR/SO/AS |
|---|---|
| **Input** : Number of epochs $K$, step-size $\alpha > 0$, and initialization $\mathbf{z}_0$ | **Input** : Number of epochs $K$, step-size $\alpha > 0$, and initialization $\mathbf{z}_0$ |
| Initialize $\mathbf{z}_0^1 \leftarrow \mathbf{z}_0$ | Initialize $\mathbf{z}_0^1 \leftarrow \mathbf{z}_0$ |
| SO: Sample $\tau \sim \text{Uniform}(\mathbb{S}_n)$ | SO: Sample $\tau \sim \text{Uniform}(\mathbb{S}_n)$ |
| **for** $k \in [K]$ **do** | **for** $k \in [K]$ **do** |
|   RR: Sample $\tau_k \sim \text{Uniform}(\mathbb{S}_n)$ |   RR: Sample $\tau_k \sim \text{Uniform}(\mathbb{S}_n)$ |
|   SO: $\tau_k \leftarrow \tau$ |   SO: $\tau_k \leftarrow \tau$ |
|   AS: Adversary chooses $\tau_k \in \mathbb{S}_n$ |   AS: Adversary chooses $\tau_k \in \mathbb{S}_n$ |
|   **for** $i \in [n]$ **do** |   **for** $i \in [n]$ **do** |
|     $\mathbf{z}_i^k \leftarrow \mathbf{z}_{i-1}^k - \alpha\omega_{\tau_k(i)}(\mathbf{z}_{i-1}^k)$ |     Solve the implicit update for $\mathbf{z}_i^k$ where, $\mathbf{z}_i^k = \mathbf{z}_{i-1}^k - \alpha\omega_{\tau_k(i)}(\mathbf{z}_i^k)$ |
|   **end** |   **end** |
|   $\mathbf{z}_0^{k+1} \leftarrow \mathbf{z}_n^k$ |   $\mathbf{z}_0^{k+1} \leftarrow \mathbf{z}_n^k$ |
| **end** | **end** |

Figure 1: GDA-RR/SO/AS and PPM-RR/SO/AS for solving (4). Violet lines denote steps that are only performed for RR, Olive lines denote the same for SO and Magenta for AS.

Problem (4) is more general than SC-SC minimax optimization, and is a special case of strongly monotone variational inequalities [13] without constraints. Notably, (4) includes the Nash Equilibrium problem for unconstrained *multiplayer* games with smooth strongly convex objectives [50] and is sometimes called a *finite-sum unconstrained variational inequality* in the literature [31]. We also highlight that smooth strongly convex *optimization* is a special case of (4). However, unlike the optimization setting, $\nu$ is no longer restricted to be the gradient of a strongly convex function, which, as we shall see, has important consequences for the attainable convergence rates of our algorithms.

### 3.2 Analysis of RR/SO

We now state the expected last-iterate convergence guarantees for Algorithms 1 and 2 for solving (4), where the expecation is taken over the stochasticity of the sampled permutation(s).

**Theorem 1** (Convergence of GDA-RR/SO and PPM-RR/SO). *Consider Problem* (4) *for the $\mu$-strongly monotone operator $\nu(\mathbf{z}) = 1/n \sum_{i=1}^n \omega_i(\mathbf{z})$ where each $\omega_i$ is l-Lipschitz, but not necessarily monotone. Let $\mathbf{z}^*$ denote the unique root of $\nu$. Then, there exists a step-size $\alpha \leq \mu/5nl^2$ for which both GDA-RR/SO and PPM-RR/SO satisfy the following for any $K \geq 1$:*

$$\mathbb{E}[|\mathbf{z}_0^{K+1}-\mathbf{z}^*|^2] \leq 2e^{-K/5\kappa^2}|\mathbf{z}_0-\mathbf{z}^*|^2 + \frac{2\mu^2 + 8\kappa^2\sigma_*^2\log^3(|\nu(\mathbf{z}_0)|n^{1/2}K/\mu)}{\mu^2 nK^2} = \tilde{O}(e^{-K/5\kappa^2}+1/nK^2),$$

*where $\kappa = l/\mu$ is the condition number and $\sigma_*^2 = 1/n \sum_{i=1}^n |\omega_i(\mathbf{z}^*)|^2$ is the gradient variance at $\mathbf{z}^*$.*

*Proof.* We present an outline for GDA-RR/SO and defer the full proof to Appendix C.2 (for GDA-RR/SO) and Appendix D.2 (for PPM-RR/SO). Furthermore, we recall that the updates of GDA-RR/SO are given by $\mathbf{z}_i^k = \mathbf{z}_{i-1}^k - \alpha\omega_{\tau_k(i)}(\mathbf{z}_{i-1}^k)$.

We begin with the following key insight from earlier works on sampling without replacement for minimization [23, 1, 37, 22]: for small enough step-sizes, the *epoch iterates* $\mathbf{z}_0^k$ of GD without replacement approximately follow the trajectory of full-batch gradient descent. To this end, we derive the following *epoch-level* update rule for GDA-RR/SO by linearizing $\omega_{\tau_k(i)}(\mathbf{z}_{i-1}^k)$ around $\mathbf{z}^*$:

$$\mathbf{z}_0^{k+1} - \mathbf{z}^* = \mathbf{H}_k(\mathbf{z}_0^k - \mathbf{z}^*) + \alpha^2\mathbf{r}_k, \tag{5}$$

where $|\mathbf{H}_k| \leq 1 - n\alpha\mu/2$ and $\mathbf{r}_k = \sum_{i=1}^{n-1}\mathbf{A}_{\tau_k(i)}\sum_{j=1}^i \omega_{\tau_k(j)}(\mathbf{z}^*)$ with $|\mathbf{A}_{\tau_k(i)}| \leq le^{1/5}$. The term $\mathbf{r}_k$ encapsulates the noise of the stochastic gradient updates accumulated over an entire epoch. To ensure convergence, we control the influence of the noise term $\mathbf{r}_k$ by using standard properties of without-replacement sample means to show that $\mathbb{E}[|\mathbf{r}_k|^2] \leq l^2n^3\sigma_*^2/4$ for both RR and SO. We then complete the proof by unrolling (5) for $K$ epochs, substituting the upper bounds for $|\mathbf{H}_k|$ and $\mathbb{E}[|\mathbf{r}_k|^2]$ wherever necessary, and setting $\alpha = \min\{\mu/5nl^2, 2\log(|\nu(\mathbf{z}_0)|\, n^{1/2}K/\mu)/\mu nK\}$.

As we show in Appendix C, the update rule (5) resembles the linearized update rule of full batch GDA with added noise. In fact, for $n = 1$, $\mathbf{r}_k = 0$ and thus, we recover the rates of full-batch GDA. Expressing GDA-RR/SO (and later AS) as noisy full-batch GDA in this fashion is a central component of our unified analysis, and relies on the fact that $\sum_{i=1}^{n} \omega_{\tau_k(i)}(\mathbf{z}^*) = 0 \ \forall \ \tau_k \in \mathbb{S}_n$, which is specific to sampling without replacement. Comparing to SGDA with replacement, we note that sampling the components i.i.d. uniformly as $u(i) \sim \text{Uniform}([n])$ gives rise to an *additional* noise term $\alpha \mathbf{p}_k$ in the update rule, where $\mathbf{p}_k = \sum_{i=1}^{n} \omega_{u(i)}(\mathbf{z}^*)$ vanishes only in expecation, and has a variance of $\mathbb{E}[|\mathbf{p}_k|^2] = n\sigma_*^2$. Subsequently, the dominant noise term for SGDA updates is $O(\alpha^2 n \sigma_*^2)$ whereas that of GDA-RR/SO is $O(\alpha^4 n^3 \sigma_*^2)$, which qualitatively demonstrates the *implicit variance reduction* of sampling without replacement. As we shall see in the complete proof, this allows RR/SO to converge faster (for large enough $K$) by carefully selecting $\alpha$. $\qquad\square$

**Comparison with lower bounds:** Since smooth strongly convex minimization is a special case of (4), the $\Omega(1/nK^2)$ lower bound established in prior works [48, 43] for smooth strongly convex minimization using GD with RR/SO also applies to GDA-RR/SO. Comparing with this lower bound, we note that the convergence rate of GDA-RR/SO is *nearly tight*, i.e., it differs from the lower bound only by an exponentially decaying term. In fact, for $K \geq 10\kappa^2 \log(n^{1/2}K)$, the convergence rate becomes $\tilde{O}(1/nK^2)$, which precisely matches the lower bound, modulo logarithmic factors.

**Comparison with uniform sampling:** Similarly, the $\Omega(1/nK)$ lower bound of SGD with replacement for smooth and strongly convex functions [44] also applies to SGDA with replacement. On the contrary, both GDA-RR and GDA-SO converge with a faster rate of $\tilde{O}(1/nK^2)$ when $K \geq 10\kappa^2 \log(n^{1/2}K)$. Thus, GDA-RR/SO provably outperform SGDA with replacement (modulo logarithmic factors) when $K \geq 10\kappa^2 \log(n^{1/2}K)$. As we show in Appendix C.2, the $\kappa^2$ dependence of this inequality cannot be improved for constant step-sizes. A similar argument also applies to stochastic PPM. To the best of our knowledge, the fastest known convergence rate for stochastic PPM is $O(1/nK)$ for minimizing smooth strongly convex functions [41]. Hence, Theorem 1 suggests that PPM-RR/SO enjoy a faster $\tilde{O}(1/nK^2)$ convergence rate *for both minimization and minimax optimization* when $K \geq 10\kappa^2 \log(n^{1/2}K)$.

### 3.3 Analysis in the Adversarial Shuffling Regime

We now consider without-replacement minimax optimization algorithms in an adversarial setting. We focus on a novel class of training-time attacks known as *data ordering attacks* proposed by Shumailov et al. [51]. These attacks differ from standard data-perturbation attacks [19] and exploit the fact that most implementations of stochastic gradient optimizers do not verify whether the permutation $\tau_k$ is truly sampled at random. Shumailov et al. [51] propose three distinct attack strategies, namely, *batch reordering*, which changes the order in which mini-batches are supplied to the algorithm, *batch reshuffling*, which changes the order in which individual data points are supplied, and *replacing* which prevents certain data points from being observed by the algorithm by consistently replacing them with other data points in the training set.

We analyze the convergence of without-replacement GDA and PPM under batch reshuffling attacks. To this end, we consider an adversarial modification of RR/SO where the permutations $\tau_k$, instead of being sampled by the algorithm, are now selected by an adversary using a strategy unknown to the algorithm. We also the assume that, while choosing $\tau_k$, the adversary is computationally unrestricted and has complete knowledge of all the components $\omega_i$, the minimax point $\mathbf{z}^*$, and the iterates $\mathbf{z}_i^k$ observed so far. We call this setup *Adversarial Shuffling* (AS) and obtain convergence rates of GDA and PPM (named GDA-AS and PPM-AS) when solving (4). Thus, our analysis naturally holds for minimization, minimax optimization as well as finite-sum multiplayer games. The details are stated in Algorithms 1 and 2, respectively. Our last iterate convergence guarantees are deterministic and hold uniformly over any sequence of permutations $\tau_1, \ldots, \tau_K$ that the adversary can choose.

**Theorem 2** (Convergence of GDA-AS and PPM-AS). *Consider Problem* (4) *for the $\mu$-strongly monotone operator $\nu(\mathbf{z}) = 1/n \sum_{i=1}^{n} \omega_i(\mathbf{z})$ where each $\omega_i$ is $l$-Lipschitz, but not necessarily monotone. Let $\mathbf{z}^*$ denote the unique root of $\nu$. Then, there exists a step-size $\alpha \leq \mu/5nl^2$ for which both GDA-AS and PPM-AS satisfy the following for any $K \geq 1$:*

$$\max_{\tau_1,\ldots,\tau_K \in \mathbb{S}_n} |\mathbf{z}_0^{K+1} - \mathbf{z}^*|^2 \leq 2e^{-K/5\kappa^2} |\mathbf{z}_0 - \mathbf{z}^*|^2 + \frac{2\mu^2 + 24\kappa^2\sigma_*^2 \log^3(|\nu(\mathbf{z}_0)|K/\mu)}{\mu^2 K^2} = \tilde{O}(e^{-K/5\kappa^2} + 1/K^2),$$

*where $\kappa, \sigma_*^2$ are as defined in Theorem 1 and $\tau_1, \ldots, \tau_K$ are the permutations chosen by the adversary.*

**Convergence rates of IG and comparison with lower bounds:** We note that the Incremental Gradient and the Incremental Proximal Point Methods, which do not shuffle the data, are a special case of GDA-AS/PPM-AS with $\tau_1, \ldots, \tau_K = id$. Thus, Theorem 2 also gives us convergence rates for GDA-IG/PPM-IG. Moreover, since GDA-AS generalizes GDA-IG and (4) covers minimization, the $\Omega(1/K^2)$ lower bound of IG established in prior works [48] for smooth strongly convex minimization also applies to GDA-AS. Thus, our obtained rate for GDA-AS is nearly tight and matches the lower bound (modulo logarithmic factors) for $K \geq 10\kappa^2 \log(K)$.

**Effectiveness of batch reshuffling:** When $K \geq 10\kappa^2 \log(K)$, $\tilde{O}(1/K^2)$ becomes the dominant term in the convergence rate of AS. This is worse than that of RR/SO by a factor of $1/n$ and causes a significant slowdown in convergence, since, in many applications, the dataset size $n$ is much larger than $K$. Thus, our analysis justifies the effectiveness of batch reshuffling attacks in reducing model accuracy and increasing training time, which is empirically verified by Shumailov et al. [51].

## 4 RR for Two-Sided PŁ Objectives

We now analyze RR for a class of smooth nonconvex-nonconcave problems where the objective $F$ satisfies a *two-sided Polyak Łojasiewicz inequality*, first proposed in Yang et al. [54]. We denote this function class as 2PŁ and formally state the assumption as follows.

**Assumption 3** (Two-sided Polyak Łojasiewicz Inequality or 2PŁ condition). *For any $\mathbf{x} \in \mathbb{R}^{d_\mathbf{x}}, \mathbf{y} \in \mathbb{R}^{d_\mathbf{y}}$, the sets $\arg\max_{\tilde{\mathbf{y}}} F(\mathbf{x}, \tilde{\mathbf{y}})$ and $\arg\min_{\tilde{\mathbf{x}}} F(\tilde{\mathbf{x}}, \mathbf{y})$ are non-empty. Furthermore, there exist positive constants $\mu_1, \mu_2$ such that $F$ satisfies the following:*

$$|\nabla_\mathbf{x} F(\mathbf{x}, \mathbf{y})|^2 \geq 2\mu_1 [F(\mathbf{x}, \mathbf{y}) - \min_{\tilde{\mathbf{x}} \in \mathbb{R}^{d_\mathbf{x}}} F(\tilde{\mathbf{x}}, \mathbf{y})], \quad |\nabla_\mathbf{y} F(\mathbf{x}, \mathbf{y})|^2 \geq 2\mu_2 [\max_{\tilde{\mathbf{y}} \in \mathbb{R}^{d_\mathbf{y}}} F(\mathbf{x}, \tilde{\mathbf{y}}) - F(\mathbf{x}, \mathbf{y})].$$

The 2PŁ condition is satisfied in several practical settings, including, but not limited to, robust least squares [12], imitation learning for linear quadratic regulators [14, 8], and various other problems in reinforcement learning and robust control [11, 8]. Clearly, any SC-SC function is 2PŁ. However, 2PŁ functions need not be SC-SC, or even convex-concave. We refer the readers to Yang et al. [54] for a detailed discussion of the 2PŁ class and its applications.

Analysis of RR for 2PŁ objectives is challenging not only due to nonconvexity-nonconcavity, but also because $F$ may not have a unique minimax point. Indeed, as we demonstrate in Appendix E.1, it is possible to construct 2PŁ functions where the set of minimax points is an unbounded proper subset of $\mathbb{R}^d$. Hence, the notion of *gradient variance at the minimax point*, which we used in our earlier analyses, is no longer meaningful. To overcome this, we impose the following assumption.

**Assumption 4** (Bounded Gradient Variance). *There exists a positive constant $\sigma$ such that the component gradient operators $\omega_i$ satisfy the following for any $\mathbf{z} \in \mathbb{R}^d$:*

$$1/n \sum_{i=1}^{n} |\omega_i(\mathbf{z}) - \nu(\mathbf{z})|^2 \leq \sigma^2.$$

### 4.1 Analysis of AGDA-RR and AGDA-AS

In order to establish the provable benefits of RR for smooth finite-sum minimax optimization of 2PŁ objectives, we propose the *Alternating Gradient Descent Ascent with Random Reshuffling* (AGDA-RR) algorithm. AGDA-RR achieves near-optimal convergence guarantees for 2PŁ objectives by combining RR with *alternating updates* [17, 57, 3] and *timescale separation* [29, 15, 16], two ideas that have been very useful for improving convergence and stability in nonconvex-nonconcave minimax optimization. Within each epoch $k \in [K]$, AGDA-RR uniformly samples a random permutation $\tau_k$, makes one full pass over the dataset, and performs gradient descent (with RR) updates for the variable $\mathbf{x}$ using the permutation $\tau_k$. This is followed by sampling another permutation $\pi_k$ and performing gradient ascent (with RR) updates for $\mathbf{y}$ using the permutation $\pi_k$. The detailed procedure is stated in Algorithm 3. We also analyze a variant of AGDA-RR in the adversarial shuffling setting, which we call AGDA-AS. The procedure, as described in Algorithm 4, is almost identical to AGDA-RR, except that the permutations $\tau_k$ and $\pi_k$ are chosen by an adversary.

| **Algorithm 3:** AGDA-RR | **Algorithm 4:** AGDA-AS |
|---|---|
| **Input** : Number of epochs $K$, step-sizes $\alpha, \beta > 0$, and initialization $(\mathbf{x}_0, \mathbf{y}_0)$ | **Input** : Number of epochs $K$, step-sizes $\alpha, \beta > 0$, and initialization $(\mathbf{x}_0, \mathbf{y}_0)$ |
| Initialize $(\mathbf{x}_0^1, \mathbf{y}_0^1) \leftarrow (\mathbf{x}_0, \mathbf{y}_0)$ | Initialize $(\mathbf{x}_0^1, \mathbf{y}_0^1) \leftarrow (\mathbf{x}_0, \mathbf{y}_0)$ |
| **for** $k \in [K]$ **do** | **for** $k \in [K]$ **do** |
|   Sample a permutation $\tau_k \in \mathbb{S}_n$ |   Adversary chooses a permutation $\tau_k \in \mathbb{S}_n$ |
|   **for** $i \in [n]$ **do** |   **for** $i \in [n]$ **do** |
|     $\mathbf{x}_i^k \leftarrow \mathbf{x}_{i-1}^k - \alpha \nabla_{\mathbf{x}} f_{\tau_k(i)}(\mathbf{x}_{i-1}^k, \mathbf{y}_0^k)$ |     $\mathbf{x}_i^k \leftarrow \mathbf{x}_{i-1}^k - \alpha \nabla_{\mathbf{x}} f_{\tau_k(i)}(\mathbf{x}_{i-1}^k, \mathbf{y}_0^k)$ |
|   **end** |   **end** |
|   Sample a permutation $\pi_k \in \mathbb{S}_n$ |   Adversary chooses a permutation $\pi_k \in \mathbb{S}_n$ |
|   **for** $i \in [n]$ **do** |   **for** $i \in [n]$ **do** |
|     $\mathbf{y}_i^k \leftarrow \mathbf{y}_{i-1}^k + \beta \nabla_{\mathbf{y}} f_{\pi_k(i)}(\mathbf{x}_n^k, \mathbf{y}_{i-1}^k)$ |     $\mathbf{y}_i^k \leftarrow \mathbf{y}_{i-1}^k + \beta \nabla_{\mathbf{y}} f_{\pi_k(i)}(\mathbf{x}_n^k, \mathbf{y}_{i-1}^k)$ |
|   **end** |   **end** |
|   $(\mathbf{x}_0^{k+1}, \mathbf{y}_0^{k+1}) \leftarrow (\mathbf{x}_n^k, \mathbf{y}_n^k)$ |   $(\mathbf{x}_0^{k+1}, \mathbf{y}_0^{k+1}) \leftarrow (\mathbf{x}_n^k, \mathbf{y}_n^k)$ |
| **end** | **end** |

Before presenting a convergence analysis, we highlight that the absence of a unique minimax point prevents us from using the squared distance to the optimum as a Lyapunov function. To this end, we use the Lyapunov function $V_\lambda : \mathbb{R}^d \to \mathbb{R}$ which was previously suggested by Yang et al. [54]. We begin by first defining the *best response function* $\Phi : \mathbb{R}^d \to \mathbb{R}$ and its minimum $\Phi^*$ as follows,

$$\Phi(\mathbf{x}) = \max_{\mathbf{y} \in \mathbb{R}^{d_{\mathbf{y}}}} F(\mathbf{x}, \mathbf{y}), \quad \Phi^* = \min_{\mathbf{x} \in \mathbb{R}^{d_{\mathbf{x}}}} \Phi(\mathbf{x}) = \min_{\mathbf{x} \in \mathbb{R}^{d_{\mathbf{x}}}} \max_{\mathbf{y} \in \mathbb{R}^{d_{\mathbf{y}}}} F(\mathbf{x}, \mathbf{y}).$$

Assumption 3 ensures that $\Phi$ is well defined and the existence of a global minimax point guarantees that $\Phi^*$ is finite. Subsequently, for any $\lambda > 0$, we define the Lyapunov function $V_\lambda$ as

$$V_\lambda(\mathbf{x}, \mathbf{y}) = [\Phi(\mathbf{x}) - \Phi^*] + \lambda[\Phi(\mathbf{x}) - F(\mathbf{x}, \mathbf{y})].$$

By definition of $\Phi$, $V_\lambda$ is non-negative for any $\lambda > 0$ and $V_\lambda(\mathbf{z}) = 0$ if and only if $\mathbf{z}$ is a minimax point of $F$. Hence, we present our convergence proofs for AGDA-RR and AGDA-AS in terms of $V_\lambda$.

**Theorem 3** (Convergence of AGDA-RR/AS). *Let Assumptions 1, 3, and 4 hold and let $\eta = 73l^2/2\mu_2^2$. Then, there exists a step-size $\alpha \leq 1/5\eta n l$ such that for $\beta = \eta\alpha$, AGDA-RR satisfies the following for $\lambda = 1/10$ and any $K \geq 1$:*

$$\mathbb{E}[V_\lambda(\mathbf{z}_0^{K+1})] \leq e^{-K/365\kappa^3} V_\lambda(\mathbf{z}_0) + \frac{\mu_1 + c\kappa^8 \sigma^2 \log^2(V_\lambda(\mathbf{z}_0) n^{1/2} K)}{\mu_1 n K^2} = \tilde{O}(e^{-K/365\kappa^3} + 1/nK^2),$$

*where $\kappa = \max\{l/\mu_1, l/\mu_2\}$ and $c > 0$ is a constant independent of $\kappa, \mu_1, \mu_2, \sigma^2$. Under the same setting, AGDA-AS satisfies the following ($\hat{c} > 0$ is a constant independent of $\kappa, \mu_1, \mu_2, \sigma^2$):*

$$\max_{\tau_1, \pi_1, \dots, \tau_K, \pi_K \in \mathbb{S}_n} V_\lambda(\mathbf{z}_0^{K+1}) \leq e^{-K/365\kappa^3} V_\lambda(\mathbf{z}_0) + \frac{\mu_1 + \hat{c}\kappa^8 \sigma^2 \log^2(V_\lambda(\mathbf{z}_0) K)}{\mu_1 K^2} = \tilde{O}(e^{-K/365\kappa^3} + 1/K^2),$$

*where $\tau_1, \pi_1 \dots, \tau_K, \pi_K$ are the permutations chosen by the adversary.*

**Convergence to a Saddle Point:** As demonstrated in Appendix E.4, the convergence guarantee of Theorem 3, which is presented in terms of $V_\lambda$, can be easily translated into an equivalent convergence guarantee in terms of $\text{dist}(\mathbf{z}, \mathcal{Z}^*)^2$, where $\mathcal{Z}^*$ denotes the set of saddle points of $F$. In particular, Theorem 3 implies the following convergence guarantee for AGDA-RR:

$$\mathbb{E}[\text{dist}(\mathbf{z}_0^{K+1}, \mathcal{Z}^*)^2] = \tilde{O}(e^{-K/365\kappa^3} + 1/nK^2),$$

as well as the following convergence rate for AGDA-AS:

$$\max_{\tau_1, \pi_1, \dots, \tau_K, \pi_K \in \mathbb{S}_n} \text{dist}(\mathbf{z}_0^{K+1}, \mathcal{Z}^*)^2 = \tilde{O}(e^{-K/365\kappa^3} + 1/K^2),$$

**Comparison with lower bounds:** Strongly convex minimization is a special case of 2PŁ minimax optimization, since minimizing the strongly convex function $f$ is equivalent to minimax optimization of the 2PŁ function $F(\mathbf{x}, \mathbf{y}) = f(\mathbf{x}) - \langle \mathbf{y}, \mathbf{y} \rangle$. In fact, the $\mathbf{x}$ iterates of AGDA-RR for $F$ are exactly

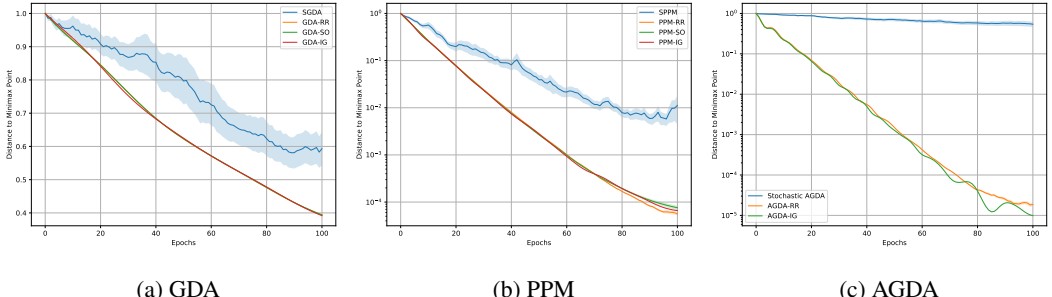

|  (a) GDA | (b) PPM | (c) AGDA |

Figure 2: Relative distance of the epoch iterates from the global minimax point (i.e. $|\mathbf{z}_0^k - \mathbf{z}^*|^2/|\mathbf{z}_0 - \mathbf{z}^*|^2$ vs $k$). The solid lines are the average over 50 runs and the shaded regions are 95% confidence intervals. The y-axis of 3a is on a linear scale whereas that of 3b and 3c is on a logarithmic scale.

that of GD with RR for $f$. Hence, the $\Omega(1/nK^2)$ lower bound for strongly convex minimization using GD with RR also applies to AGDA-RR. Thus our convergence rate for AGDA-RR is nearly tight and matches the lower bound (modulo logarithmic factors) for $K \geq 730\kappa^3 \log(n^{1/2}K)$. Similarly, the Incremental Gradient version of AGDA is a special case of AGDA-AS with $\tau_1, \pi_1, \ldots, \tau_K, \pi_K = id$ and hence, AGDA-AS is nearly tight and matches the $\Omega(1/K^2)$ lower bound (modulo logarithmic factors) for $K \geq 730\kappa^3 \log(K)$.

**Comparison with stochastic AGDA:** Similarly, the $\Omega(1/nK)$ lower bound of SGD with replacement also holds for the Stochastic AGDA algorithm [54], which samples the component functions with replacement and performs two-timescale alternating updates similar to AGDA-RR. Hence, Theorem 3 demonstrates that AGDA-RR provably outperforms stochastic AGDA when $K \geq 730\kappa^3 \log(n^{1/2}K)$.

**Bounded iterate assumption** Assumption 4 is also used in analyzing RR for PŁ function minimization [34]. In this setting, an alternative *bounded iterate assumption*, which assumes that all the iterates $\mathbf{z}_i^k$ lie within a compact set, has also been used [1]. As shown in Appendix E.2, our proof of Theorem 3 easily adapts to this assumption. In the absence of either assumption, Li et al. [28] use time-varying step-sizes to obtain *asymptotic* $O(1/K^2)$ rates for RR on PŁ (and more generally for KŁ) minimization.

## 5 Experiments

We evaluate our theoretical results by benchmarking on finite-sum SC-SC quadratic minimax games. This class of problems appears in several applications such as reinforcement learning [11], robust regression, [12] and online learning [26]. The objective $F$ and the components $f_i$ are given by:

$$F(\mathbf{x}, \mathbf{y}) = 1/n \sum_{i=1}^n f_i(\mathbf{x}, \mathbf{y}) = \frac{1}{2}\mathbf{x}^T \mathbf{A}\mathbf{x} + \mathbf{x}^T \mathbf{B}\mathbf{y} - \frac{1}{2}\mathbf{y}^T \mathbf{C}\mathbf{y},$$

$$f_i(\mathbf{x}, \mathbf{y}) = \frac{1}{2}\mathbf{x}^T \mathbf{A}_i\mathbf{x} + \mathbf{x}^T \mathbf{B}_i\mathbf{y} - \frac{1}{2}\mathbf{y}^T \mathbf{C}_i\mathbf{y} - \mathbf{u}_i^T \mathbf{x} - \mathbf{v}_i^T \mathbf{y},$$

where $\mathbf{A}$ and $\mathbf{C}$ are strictly positive definite. We generate the components $f_i$ randomly, such that $\sum_{i=1}^n \mathbf{u}_i = \sum_{i=1}^n \mathbf{v}_i = 0$ and the expected singular values of $\mathbf{B}$ are larger than that of $\mathbf{A}$ and $\mathbf{C}$. This ensures that the bilinear coupling term $\mathbf{x}^T \mathbf{B}\mathbf{y}$ is sufficiently strong, since a weak coupling practically reduces to quadratic minimization, which has already been investigated in prior works. Finally, to investigate how the presence of nonconvex-nonconcave components impacts convergence, a few randomly chosen $f_i$'s are allowed to be nonconvex-nonconcave quadratics. For each algorithm analyzed in the text, we benchmark sampling without replacement against uniform sampling by running each method for 100 epochs using constant step-sizes that are selected independently for each method via grid search. Further details regarding the setup is discussed in Appendix G.

We present our results in Figure 2, where we plot the relative distance of the epoch iterates from the minimax point, defined as $|\mathbf{z}_0^k - \mathbf{z}^*|^2/|\mathbf{z}_0 - \mathbf{z}^*|^2$, averaged over 50 independent runs. In agreement with our theoretical findings, sampling without replacement consistently outperforms uniform sampling

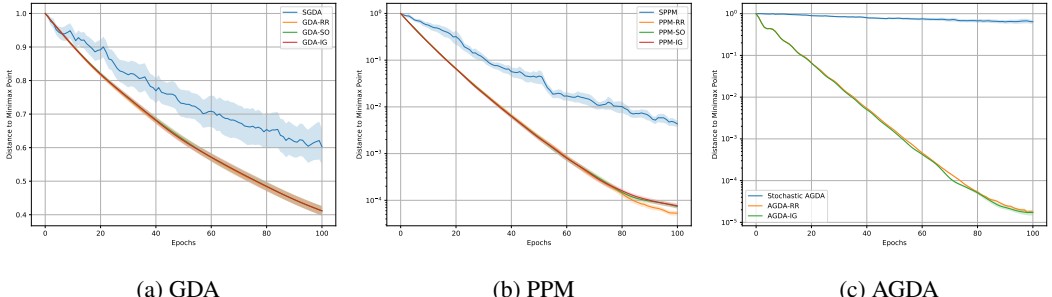

|   (a) GDA   |   (b) PPM   |   (c) AGDA   |

Figure 3: Convergence of GDA, PPM and AGDA averaged over 20 random instances. Shaded regions represent 95% confidence intervals.

across all three setups. Furthermore, to demonstrate that our observations are not particular to one specific instance, we repeat the experiment for 20 independently sampled quadratic games, and for each instance, perform 5 independent runs of each algorithm and plot the average relative distance of the epoch iterates from the minimax point. The results, presented in Figure 3, substantiates the superior convergence of sampling without replacement across multiple instances.

## 6 Conclusion

We derived near optimal convergence rates for several without-replacement stochastic gradient algorithms for finite-sum minimax optimization, and demonstrated that they converge faster than algorithms that use uniform sampling. We considered two problem classes, strongly convex-strongly concave problems (generalized to unconstrained strongly monotone variational inequalities) and nonconvex-nonconcave problems with two-sided PŁ objectives. We also formally defined *adversarial shuffling*, where an attacker can control the order in which data points are supplied to the optimizer, and analyzed minimax optimization in this regime. Interesting future directions include the analysis of inexact proximal point methods, more general function classes, and time-varying step-sizes.

## Acknowledgements and Disclosure of Funding

Michael Muehlebach and Bernhard Schölkopf thank the German Research Foundation and the Branco Weiss Fellowship, administered by ETH Zurich, for the generous support.

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
