# OpenReview forum: "Sampling without Replacement Leads to Faster Rates in Finite-Sum Minimax Optimization"
_NeurIPS.cc/2022/Conference — NeurIPS 2022 Accept_

### Official Review · Reviewer_R12f · 2022-07-10

**Rating:** 6
**Confidence:** 3
**Soundness:** 3 good
**Presentation:** 3 good
**Contribution:** 3 good

**Summary:**

This paper considers finite sum minimax optimization problems. The authors propose to use gradient descent ascent/proximal point method with RR and AS data sampling schemes. They obtain iteration complexity results for both strongly convex-strongly concave or 2PL functions, matching the state-of-the-art ones for minimization problems.

**Questions:**

I have the following major concerns

1. By checking the proof of Theorem 1, I can understand that the main difficulty (or let's say the main difference to the existing analysis of RR) lies in rewriting the GDA-RR update as a GDA form. The authors may highlight this difficulty clearly in the main context to show the technical difficulty as other parts are really similar to the existing RR analysis.

2. Lemma B.3 is quite important for establishing the $O(1/nK^2)$ result. It fully utilizes the randomness of random shuffling in terms of expectation. However, such a lemma (as I know) was already used in [1, Lemma 1]. I would highly recommend the authors clearly state this reference in their Lemma B.3. Citing this existing lemma rather than reinventing it is also welcome.

3. Is it possible to remove the bounded variance assumption in 2PL case? It seems that this assumption is removed for random shuffling algorithm in KL inequality setting (more general than PL as I know) in a recent paper [5].

4. In 2PL setting, is it possible to use dist$(z,Z^*)$ as the Lyapunov function, where $Z^*$ is the set of saddle points? If not, what is the underlying difficulty?

5. The authors should give a more comprehensive literature review of the RR algorithm. Some immediately related references: 1) The very pioneering papers [2]-[3].  2) [4], which studies RR with momentum. 3) [5], which studies RR in KL inequality setting. 4) [6], which shows the limitation of RR under bad conditioning. 5) [7], which applies to federated learning setting. More related works are welcome. It is important to put this paper in the correct position in the literature. Potential comparisons between this paper and the existing literature will also be appreciated.

[1] Mishchenko, K., Khaled, A., & Richtárik, P. (2020). Random reshuffling: Simple analysis with vast improvements. Advances in Neural Information Processing Systems, 33, 17309-17320.

[2] Recht, B., & Ré, C. (2012, June). Toward a noncommutative arithmetic-geometric mean inequality: Conjectures, case-studies, and consequences. In Conference on Learning Theory (pp. 11-1). JMLR Workshop and Conference Proceedings.

[3] Gürbüzbalaban, M., Ozdaglar, A., & Parrilo, P. (2015). Why Random Reshuffling Beats Stochastic Gradient Descent. arXiv preprint arXiv:1510.08560.

[4] Tran, T. H., Nguyen, L. M., & Tran-Dinh, Q. (2020). SMG: A Shuffling Gradient-Based Method with Momentum. arXiv preprint arXiv:2011.11884.

[5] Li, X., Milzarek, A., & Qiu, J. (2021). Convergence of random reshuffling under the kurdyka-{\L} ojasiewicz inequality. arXiv preprint arXiv:2110.04926.

[6] Safran, I., & Shamir, O. (2021). Random shuffling beats SGD only after many epochs on ill-conditioned problems. Advances in Neural Information Processing Systems, 34, 15151-15161.

[7] Mishchenko, K., Khaled, A., & Richtárik, P. (2021). Proximal and federated random reshuffling. arXiv preprint arXiv:2102.06704.

**Ethics Review Area:**

["I don’t know"]

**Limitations:**

Yes.

**Strengths And Weaknesses:**

-Strength
This paper provides (nearly) optimal expected iteration complexity bounds for both strongly convex-strongly concave or 2PL functions.

-Weakness
See comments below.

---

> ### Author Response · Authors · 2022-08-02
> **On Highlighting the Key Difficulty of Our Analysis, Lemma B.3 and Literature Survey**
>
> Thank you for your detailed review, positive evaluation of our contributions, and for your valuable suggestions which have greatly helped us in improving our manuscript. We hope your concerns are addressed in our response below.
>
> # Highlighting the key difficulty in our analysis
>
> Thank you for this suggestion. We agree that one of the key challenges in our analysis lies in expressing the epoch level update rule of GDA without replacement (and PPM) in a form that resembles the linearized update rule of full batch GDA (and PPM, respectively). In fact, expressing the epoch level update in this form is key to developing a general proof strategy that simultaneously handles RR, SO and AS with little to no modification. We have updated our manuscript to highlight this point in the proof sketch of Theorem 1 (Lines 188-191). We have also discussed this point in greater detail in Theorem C.2 of the Appendix (complete unified proof of GDA-RR/SO, Lines 731-742), highlighting connections to the insights developed in some of the foundational works that analyze RR for minimization.
>
> # Lemma B.3
>
> Thank you for this suggestion. We have updated our manuscript to state that Lemma B.3 has been previously used in [1] to analyze RR/SO for minimization (which we consider to be an important contribution) and have also referenced [1, Lemma 1] in the statement of the Lemma. However, we believe that Lemma B.3 / [1, Lemma 1] by itself is a relatively standard result in statistics on the variance of random sampling without replacement [2, 3].
>
> # Literature Survey
>
> Thank you for the references. We have added a comprehensive literature review of RR, SO and IG in Appendix F. Since camera-ready versions of accepted papers are allowed to add an extra page, we would be happy to incorporate this section in the main paper if our manuscript is accepted for publication.
>
> - [1] K. Mishchenko, A. Khaled, P. Richtarik (2020) Random Reshuffling: Simple Analysis with Vast Improvements. Neural Information Processing Systems, 2020.
>
> - [2] W. Cochran (1977) Sampling Techniques. Wiley.
>
> - [3] J. Rice (1988) Mathematical Statistics and Data Analysis. Wadsworth

---

> ### Author Response · Authors · 2022-08-02
> **On Distance to the Set of Saddle Points as a Lyapunov Function and Removing Bounded Gradient Variance Assumption**
>
> # On the Possibility of Using $\textrm{dist}(\mathbf{z}, \mathcal{Z}^*)$ as a Lyapunov Function
>
> The choice of $V_{\lambda}$ as a Lyapunov function is motivated by the fact that it is amenable to the derivation of a descent lemma using the noisy epoch level updates of AGDA-RR/AS, which can then be unrolled to obtain a convergence guarantee. We believe the difficulty in using $\textrm{dist}(\mathbf{z}, \mathcal{Z}^*)$ as a Lyapunov function lies in the fact that deriving an equivalent descent lemma for this function using the noisy epoch-level update rule is, in our experience, considerably more involved (and might also not be possible). This situation parallels that of the convergence analysis of GD/SGD for PL function minimization, where the function gap $f(x) - f^*$ is typically used as a Lyapunov function [1, 2].
>
> However, we would like to highlight that it is possible to convert our convergence rates presented in terms of $V_{\lambda}$ into an equivalent convergence rate in terms of $\textrm{dist}(\mathbf{z}, \mathcal{Z}^*)^2$. In particular, using the properties of 2PL functions, one can relate $V_{\lambda}(\mathbf{z})$ to $\textrm{dist}(\mathbf{z}, \mathcal{Z}^*)^2$ as follows:
>
> $ \textrm{dist}(\mathbf{z}, \mathcal{Z}^*)^2 \leq \max [ \frac{2}{\mu_1}(\frac{L^2}{2 \mu_2^2} + 1), \frac{4}{\lambda \mu_2} ] V_{\lambda}(\mathbf{z}).$
>
> We have updated our manuscript to present a complete proof of this relation in Appendix E.4. In conjunction with our results in Theorem 3, the obtained inequality implies that AGDA-RR satisfies a convergence guarantee of the form $\mathbb{E}[\textrm{dist}(\mathbf{z}^K_0, \mathcal{Z}^*)^2] = \tilde{O}(\exp(\frac{-K}{365 \kappa^3}) + \frac{1}{nK^2})$, while AGDA-AS satisfies $ \max_{\tau_1, \pi_1, \ldots, \tau_K, \pi_K \in \mathbb{S}_n} \textrm{dist}(\mathbf{z}^K_0, \mathcal{Z}^*)^2 = \tilde{O}(\exp(\frac{-K}{365 \kappa^3}) + \frac{1}{K^2})$. A detailed discussion on this subject has been performed in Appendix E.4.
>
> # On Removing the Bounded Variance Assumption
> Thank you for bringing [3] to our notice. To the best of our understanding, [3, Theorem 3.10] uses a Chung's Lemma-style result (namely, [3, Lemma 3.9]) to obtain an *asymptotic* convergence rate of $O(1/K^2)$ for PL functions (in terms of the squared distance of the epoch iterates from the set of minimizers) without assuming bounded gradient variance or bounded iterates. (The result is asymptotic since [3, Theorem 3.10] holds only when the number of epochs $K$ is "sufficiently large" and it is not explicitly quantified how large $K$ needs to be for the result to hold). We have updated Lines 298-303 of our manuscript to acknowledge this work and have also referenced it in our literature review (Appendix F).  Please note that the asymptotic rate presented in [3] is $O(1/K^2)$ and not $O(1/nK^2)$ since [3, Theorem 3.10] does not quantify the dependence of the convergence rate on $n$.
>
> As such, we conjecture that incorporating carefully chosen time-varying step sizes into our analysis of 2PL functions would also allow us to remove the bounded gradient variance assumption, by means of relatively standard techniques in stochastic approximation (e.g. some variant of Chung's Lemma [4]). However, we believe doing so will add an additional layer of complexity, which has the potential drawback of distracting the reader from the key points of our analysis (namely, obtaining an epoch level update rule for AGDA-RR that resembles that of full-batch AGDA with added noise, controlling the influence of the noise in expectation using the variance of without-replacement sample averages, and finally performing a Lyapunov analysis for this noisy update rule). To this end, while the analysis of time-varying step sizes and the removal of the bounded variance assumption is an important contribution and an interesting avenue for future work, we believe it is outside of the scope of the current manuscript.
>
> - [1] H. Karimi, J. Nutini, M. Schmidt (2016). Linear Convergence of Gradient and Proximal-Gradient Methods Under the Polyak-{\L}ojasiewicz Condition. European Conference on Machine Learning 2016.
>
> - [2] A Wilson (2018). Lyapunov Arguments in Optimization. PhD thesis, University of California, Berkeley, 2018.
>
> - [3] X. Li, A. Milzarek, J. Qiu (2021). Convergence of random reshuffling under the Kurdyka-{\L}ojasiewicz inequality. arXiv preprint arXiv:2110.04926.
>
> - [4] K.L. Chung (1954). On A Stochastic Approximation Method. The Annals of Mathematical Statistics, 1954.

---

> > ### Comment · Reviewer_R12f · 2022-08-06
> > **Thanks for the response.**
> >
> > Most of my concerns are addressed properly. In terms of removing the bounded variance assumption, I meant that the authors might mimic the techniques of [3] to establish the algorithmic recursion of the proposed algorithms, i.e., show something like [3, Lemma 3.2]. What the current paper is going to establish is complexity bound. It should not be necessary to use Chung's lemma. Instead, once you get something like [3, Lemma 3.2], the complexity of $O(1/nK^2)$ in the sense of expectation directly follows by using proper time-varying step sizes. Thus, it seems that the authors do not need to worry about the applicability of any technique after [3, Lemma 3.2].
> >
> > Overall, I think this paper can be helpful for RR and Minimax community. However, as recommended by the scoring system, I would say 6 is appropriate as this paper has solid techniques and might have moderate-to-high impact.

---

### Official Review · Reviewer_4yKG · 2022-07-11

**Rating:** 6
**Confidence:** 3
**Soundness:** 4 excellent
**Presentation:** 4 excellent
**Contribution:** 3 good

**Summary:**

The authors derive convergence rates for stochastic gradient algorithms for finite-sum minimax optimization *without replacement*. They consider both (1) the smooth and strongly convex-strongly concave setting, and (2) 2-sided Polyak-Lojasiewicz inequality setting. The rates are better than the convergence rates for the with-replacement algorithms and match known lower bounds (up to logarithmic factors) when the epoch number K is large (as a function of the condition number). Specifically, the authors show the following.

* For convex-concave objectives, gradient descent ascent (GDA) as well as the proximal point method (PPM) achieve rates of $\tilde O(\sigma_*^2/(nK^2))$ for $K=\Omega(\kappa^2)$, where $K$ is the number of epochs, $\sigma_*^2$ is the gradient variance, $n$ is the number of terms in the sum, and $\kappa$ is the condition number.
* For 2-sided PL objectives, similar results hold for $K=\Omega(\kappa^3)$.
* The convergence rates are slowed by a factor of $n$ to $\tilde O(\sigma^2/K^2)$ when the data is adversarially shuffled, which is tight (up to log factors).

The proofs proceed by linearizing the update around the minimax point, then controlling the noise term using the variance of without-replacement sample means.


**Questions:**

Given that the same convergence rate is obtained for RR/SO, is there a reason to prefer one over the other?

Why do 2-sided PL objectives necessitate a different algorithm from convex-concave objectives?

The axes in Figure 3 are hard to read.


**Limitations:**

Yes.

**Strengths And Weaknesses:**

The authors present a unified analysis of stochastic gradient algorithms for finite-sum minimax optimization without replacement, that works for many variants (GDA vs. PPM, random reshuffling vs. shuffle once vs. adversarial shuffling). The rates indeed show the benefit of sampling without replacement. The extension to adversarial shuffling is particularly interesting as it quantifies the effect of a data-ordering attack. The paper is clear and well-organized.

The results are not particularly surprising since they parallel the existing results for optimization, though it remains valuable to work out the results for minimax optimization.

---

> ### Author Response · Authors · 2022-08-02
> **Response to Reviewer 4yKG**
>
> Thank you for your positive evaluation of our work. We are glad you find our analysis of adversarial shuffling interesting. We hope our response below addresses your concerns.
>
> # On RR vs SO for Strongly Convex-Strongly Concave Minimax Optimization
>
> Our analysis demonstrates that for the class of smooth strongly convex-strongly concave minimax problems (or more broadly, smooth strongly monotone variational inequality problems), RR and SO have the same convergence rate, and as such, there is no apparent reason to prefer one over the other for this specific class. This conclusion is also in agreement with existing literature in the minimization setting [1], which shows that RR and SO exhibit similar performance in smooth strongly convex minimization.
>
> # Necessity of a Different Algorithm for Two-Sided PL Objectives
>
> The choice of two-timescale alternating updates for two-sided PL (or 2PL) objectives is primarily motivated by the fact that, even for the deterministic minimax problem, it is not known whether the simultaneous GDA algorithm can achieve provable global convergence for 2PL objectives, whereas two-timescale alternating GDA is known to exhibit global linear convergence [2]. Additional motivation for this choice is also rooted in the fact that timescale separation and alternating updates are known to promote convergence and stability in minimax optimization [3, 4].
>
> - [1] K. Mishchenko, A. Khaled, P. Richtarik (2020) Random Reshuffling: Simple Analysis with Vast Improvements. Neural Information Processing Systems, 2020.
>
> - [2] J. Yang, N. Kiyavash, N. He (2020) Global Convergence and Variance Reduction for a Class of Nonconvex-Nonconcave Minimax Problems. Neural Information Processing Systems, 2020.
>
> - [3] G. Gidel, R. A. Hemmat, M. Pezeshki, R. L. Priol, G. Huang, S. Lacoste-Julien, and I. Mitliagkas (2019) Negative momentum for improved game dynamics. International Conference on Artificial Intelligence and Statistics, 2019.
>
> - [4] T. Lin, C. Jin, and M. I. Jordan (2020). On gradient descent ascent for nonconvex-concave minimax problems. International Conference on Machine Learning, 2020.

---

> > ### Comment · Reviewer_4yKG · 2022-08-08
> > **Thanks for the response.**
> >
> > I thank the authors for answering my questions. I will keep my score.

---

### Official Review · Reviewer_b3Wj · 2022-07-11

**Rating:** 4
**Confidence:** 3
**Soundness:** 3 good
**Presentation:** 3 good
**Contribution:** 2 fair

**Summary:**

The paper shows the convergence of stochastic GDA with random reshuffling (RR), shuffle once (SO) and adversarial shuffling (AS) for strongly-convex-strongly-concave min-max problems. It also extends to the two-sided PL condition with alternating GDA.

**Questions:**

Questions:

(a) PPM has an implicit step. Extragradient and PPM are always considered closely related. Is it possibly to extend to extragradient with RR? Usually PPM allows large stepsize (arbitrary large in deterministic setting) for convex optimization. Does it allow larger stepsize for PPM here?

(b) What is the technical novelty compared to RR in minimization?

minor comments:

(a) I suggest to include $\kappa$ for the 1/K^2 terms in the contribution part.

(b) I suggest to use $\Vert \cdot \Vert$ for norm


**Strengths And Weaknesses:**

strength: the paper is easy to follow and well-organized.

weakness: the techniques for RR has already been established for strongly-convex optimization. It is not surprising to extend it to strongly-convex-strongly-concave minmax problems.

---

> ### Author Response · Authors · 2022-08-02
> **Regarding Novelty and Contributions**
>
> Thank you for taking the time to review our work. We are glad you find our presentation well-organized. We hope the following points are able to address your concerns regarding the technical novelty and contributions of our work.
>
> 1. **Generality**: To the best of our understanding, the scope of our results is much more general than that of existing works on RR/SO for strongly convex minimization. Since our framework analyzes the broad class of strongly monotone Variational Inequality (or VI) problems, our results not only cover strongly convex minimization and strongly convex-strongly concave minimax problems, but also include problems such as multiplayer games with strongly convex cost functions. In addition, we believe that our proof techniques, which allow for a unified treatment of RR, SO and AS for strongly monotone VI problems without relying on any complex mathematical machinery (since the only tools that we employ are elementary properties of smooth functions and the variance of without-replacement sample averages), are an important contribution due to their high level of generality and accessibility.
>
> 2. **Analysis of PPM**: Even in the context of minimization, our unified analysis of PPM-RR/SO is a novel contribution, since, to the best of our knowledge, our result is the first to establish that PPM-RR/SO can exhibit faster $O(1/nK^2)$ convergence than Stochastic PPM (with uniform sampling) for finite-sum smooth strongly convex minimization problems (and more generally for finite-sum smooth strongly monotone variational inequalities). Furthermore, as a consequence of our analysis of PPM-AS, we also obtain a non-asymptotic $\tilde{O}(1/K^2 + \exp(-K/5\kappa^2))$ convergence rate for the Incremental Proximal Point Method.
>
> 3. **Data Ordering Attacks**: To the best of our knowledge, our work is the first to explicitly quantify the effect of data ordering attacks on the convergence of optimization algorithms, by means of our analysis in the adversarial shuffling regime. This has also been noted as an interesting contribution by Reviewer KJCr and Reviewer 4yKG.
>
> 4. Lastly, we believe it is not self-evident that results from strongly convex minimization can be extended to strongly convex-strongly concave minimax optimization (or more generally, to strongly monotone variational inequalities). To this end, we believe, demonstrating that RR/SO can outperform uniform sampling for strongly monotone VI problems is a valuable contribution, which is also recognized by Reviewer 4yKG. Furthermore, our work goes beyond the strongly convex-strongly concave regime by presenting guarantees for RR on a class of nonconvex-nonconcave problems. In addition, our analysis is general enough to capture adversarial shuffling with little to no modification.

---

> > ### Comment · Reviewer_b3Wj · 2022-08-08
> > **Technical novelty**
> >
> > Thanks for the authors' response!
> >
> > I am not fully convinced about the technical novelty here. As Reviewer R12f pointed out,  "the main difficulty (or let's say the main difference to the existing analysis of RR) lies in rewriting the GDA-RR update as a GDA form, other parts are really similar to the existing RR analysis.". However, the convergence rate for GDA in Theorem C.1 is quite standard in the literature, either by the book "F. Facchinei and J.-S. Pang. Finite-dimensional variational inequalities and complementarity problems." or many recent literature. The proof of Theorem C.1 define some matrices M_k which makes the proof looks more involving, but actually in the end the norm of the related operator can be easily bounded (the proof can also be written in a way without defining M_k). Also, for the proof of Theorem C.2 looks complicated by defining matrices J_k and M_k, but in the end they can be easily bounded with Lipschitz.
> >
> > I do not believe that the proof of GDA-RR is necessarily more challenging than GD-RR in the minimization case. For example, look at the reference "J. Haochen and S. Sra. Random shuffling beats SGD after finite epochs", in its arXiv version, they also need to approximate RR with GD update. As far as I see, their proof can be easily modified to accommodate minmax or VI setting. For example, in the equation (A.1) of their appendix, it suffices to replace the inequality (that is specific to minimization) to the inequality using strongly monotone (this is the also key to why GD has kappa dependency in minimizaton, but kappa^2 in minmax). Then I believe the rest part of it can be derived similarly.

---

> ### Author Response · Authors · 2022-08-02
> **Other Concerns**
>
> # Analysis of Extragradient without replacement
>
> The linearization technique that we use for analyzing without-replacement GDA/PPM is general, and as such, can be adapted to any approximate proximal point method such as Extragradient (EG). For the particular case of the EG update rule, $z^k_{i+1} = z^k_i - \alpha \omega_{\tau_k(i)}(y^k_{i+1})$  where $y^k_{i+1} = z^k_{i} - \alpha \omega_{\tau_k(i)}(z^k_i)$, one possible approach could be to first linearize $y^k_{i+1}$ about $z^*$, plug in the obtained linearization into the expression for $z^{k}_{i+1}$, and linearize the resultant once again about $z^*$. One could then obtain a linearized epoch-level update rule for EG-RR/SO/AS and perform a unified analysis in a manner similar to Theorem 1. The analysis of without-replacement approximate proximal point methods is an interesting avenue for future work which could benefit from the techniques highlighted in our manuscript.
>
> # Step-sizes for PPM-RR/SO
>
> Unlike prior works that analyze stochastic proximal point methods for minimization [1, 2], our analysis allows the components $f_i$ to be arbitrary smooth nonconvex-nonconcave functions. To facilitate this increased generality, our analysis requires us to control the influence of the noise terms contributed by the nonconvex-nonconcave components (in both $\mathbf{H}_k$ and $\mathbf{r}_k$) by appropriately tuning the step sizes. As such, we conjecture that imposing further restrictive assumptions on the component functions $f_i$ (such as strong convexity-concavity) may allow us to use larger step sizes in our analysis.
>
> - [1] A. Patrascu, I. Necoara (2018) Nonasymptotic convergence of stochastic proximal point methods for constrained convex optimization. Journal of Machine Learning Research, 2018
>
> - [2] E. Ryu, S. Boyd (2016) Stochastic proximal iteration: A non-asymptotic improvement upon
> stochastic gradient descent. https://web.stanford.edu/~boyd/papers/spi.html.

---

### Official Review · Reviewer_KJCr · 2022-07-15

**Rating:** 7
**Confidence:** 3
**Soundness:** 3 good
**Presentation:** 3 good
**Contribution:** 3 good

**Summary:**

Authors analyze variants of stochastic gradient descent ascent (SGDA) methods without replacement to solve minimax first order optimization. Authors claim that, despite all the studies about stochastic methods with replacement, enforcing a pass over the whole set of data at each epoch is a better choice. This has first been shown empirically, and the reason why methods with replacement have been the center of interest is because of the underlying assumptions that ease the theoretical study.
Therefore, methods without replacement lately received some interest, and first theoretical results arose for minimization problem.
This paper proposes a proof of the gain of speed of SGDA's convergence on minimax problems in the "no replacement" regime.

The different studied variations of the algorithm are:
- the way to order data: reshuffling at each epoch, shuffling once for all, or ordering arbitrarily removing the stochasticity aspect of the algorithm;
- the way to perform descent and ascent: simultaneously or not.

There are also 2 different studied setting. In both of them, the objective function (expressed as an average of "component" functions) is assumed smooth. The difference between the 2 settings lies in the assumption made on the component functions:
- first, assuming that all component functions are strongly convex- strongly concave;
- then, assuming that they verify a 2 sided PL inequality instead.

Additional assumptions are one of these:
- bounded variance of gradients at optimum;
- uniformly bounded variance of gradients.

Authors claim bounds improvements with respect to the "with replacement" case, and tight guarantees in their setting.

**Questions:**

- l.75: I think it is worth adding the word "smooth" as well. This is present in introduction, but it might be worth repeating it here in boldface as for the other assumptions. (Same in line 89)
- l.199: typo "comparison".
- l.254: typo "regulators".
- l.674: here, we need to add the assumption of twice differentiability.
- l.675: typo in index when defining $J_{\tau_k(i)}$. The one that is defined might be useful for the proximal based method, but not the one studied here.
- l.709: I suggest an intermediate line of computation here. Especially since the counter $j$ is reused for another sum. After first equality, the product is developed into a sum over some new counter $l$, then we can commute sum over $j$ and sum over $l$, and finally $j$ becomes $t_1$ merging the 2 last sums. But $l$ has been renamed $j$ which can be confusing.
- l.714: there is a missing $n$ in the LHS.
- l.719: There are a lot of loose inequalities here. Isn't there any way to take advantage of tighter inequalities? Getting a weaker constraint on $\alpha$ for instance. It seems that the only gain will be constant. But isn't it interesting optimizing them?
- l.740: typo: the sum counter must go up to $n$.

**Limitations:**

The only limitations are the assumptions which are clearly stated.

**Strengths And Weaknesses:**

Strengths:
- this paper brings new results of convergence and theoretical support of observed phenomenons.
- Moreover, guarantees are obtained under very weak assumptions and for various variants, even in adversarial setting.
- Finally, results are well presented and literature review is extensive although I cannot be sure if exhaustive as I am not very familiar with minimax related literature.

Weakness:
I noticed unclear statements related on O usage. This notation has a precise mathematical meaning, namely $a_n = O(b_n)$ when there exists a constant $C$ such that $a_n \leq C b_n$ holds for all $n$. Therefore, we have to be clear which variables are varying and which are fixed. In this paper, several variables are introduced and guaranteed bounds depend on $l$, $\mu$, $n$, $K$, $\sigma$ (or $\sigma_*$) and $\|z_0 - z^* \|$. One has to be clear using the O notation which of these are considered as fixed by the problem and which are varying in the O.
For instance, when looking at results of Thm 1 which are summed up in line 79 of the introduction, since $\mu, l$ disappeared from the second term and $\|z_0 - z^*\|$ from the first one, we can conclude that they are considered as constants of the problem, that are fixed at first and only the other variables can vary up to infinity.
To be clear, I also assume that $\sigma$ is fixed when I read some O(1/nK^2) in line 68 for example.
But I cannot know from this article whether we are interested by asymptotical behavior of the rate when $K$ tends to infinity, or also when $n$ grows:
- If $n$ is a fixed constant of the problem, therefore, O(1/nK^2) = O(1/K^2) and all the discussion about having this $n$ or not depending on the adversarial setting or not has to be done on the accurate bound, not using the O notation.
- If we are interested in the variation of $n$, I don't agree on the fact that $exp(-K / 5\kappa^2) + \sigma_*^2/nK^2 = O(1/nK^2)$. Indeed, if we fix $K$ and make $n$ tends to infinity, the RHS tends to 0 but not the LHS, which makes the domination of the LHS by the RHS impossible. Authors specifically asked that $K = \Omega(\kappa^2)$, but first, this assumption does not have any impact on the O notation as $\kappa$ is fixed, and moreover, we can rewrite $exp(-K / 5\kappa^2) + \sigma_*^2/nK^2 = 1/K^2 [ K^2 exp(-K / 5\kappa^2) + \sigma_*^2/n ]$ and bound the term into brackets. Its maximum is reached for $K = 10 \kappa^2$ and leads to $100\kappa^4 exp(-2) + \sigma_*^2/n $, which is not a $O(1/n)$ if $n$ is not a $O(1)$. This is a $O(1)$ and leads to a $O(1/K^2)$ rate, not $O(1/nK^2)$.
And this can never change with additional assumption of the form $K = \Omega(\kappa^{\{\text{some exponent}\}})$. One would need to enforce a relation between $K$ and $n$ in order to expect an improvement. For example, with $K \geq 5\kappa^2 (1 + \varepsilon) log(n)$ for any $\varepsilon>0$, the statement comes true.

---

> ### Author Response · Authors · 2022-08-02
> **Regarding O Notation and Epoch Requirement on K**
>
> Thank you for your detailed and thoughtful review, and for your positive evaluation of our work. Your insightful suggestions have greatly helped us in improving our manuscript. We hope our response is able to address your concerns.
>
> # Regarding Use of $O$ and $\tilde{O}$ Notation
>
> We apologize for the lack of clarity regarding the $O$ notation in some parts of the text and appreciate your helpful feedback on the same. We clarify that we are interested in the behavior of the convergence rate as both $n$ and $K$ grow, and treat parameters such as $\kappa$, $\mu$, $\sigma$ (or $\sigma^*$) and $|z_0 - z^*|$ as constant factors (hereafter called problem-specific constants). We have updated our manuscript to describe our usage of the $O$ and $\tilde{O}$ notation in Section 2 Lines 126-128, where we explicitly state that our usage of the $O$ characterizes the dependence of our rates on $n$ and $K$ while suppressing constants such as $\kappa, \mu, \sigma$, etc. (and $\tilde{O}$ also suppresses logarithmic factors of $n$ and $K$). To ensure consistency, we have suppressed said problem-specific constants in every occurrence of the $O, \tilde{O}$ and $\Omega$ notation in our manuscript. Furthermore, to ensure that the article still precisely quantifies the dependence of our convergence rates on $\kappa, \mu, \sigma^2$, etc. (as well as logarithmic factors of $n$ and $K$), the statement of all our theorems now state our obtained convergence rates both with and without the $\tilde{O}$ notation
>
> # Regarding the use of $K = \Omega(\kappa^2)$
>
> Thank you for pointing this out. We agree with your statement that, in order to ensure $\exp (-K/5\kappa^2) + 1/nK^2 = O(1/nK^2)$, one needs to assume $K \geq 10 \kappa^2 \log (n^{1/2}K)$. Similarly, for GDA/PPM-AS, one needs to assume $K \geq 10 \kappa^2 \log (K)$ to obtain a $\tilde{O}(1/K^2)$ rate. We have updated our manuscript to reflect this appropriately by replacing every occurrence of $K \geq \Omega(\kappa^c)$ with precise epoch requirements of the form $K \geq C \kappa^{a} \log(n^{b}K)$ for some constants $C, a, b > 0$ (specific values depending on the algorithm under analysis).
> We also highlight that, in the absence of structural assumptions (like convexity) on the components $f_i$, several prior results on RR/SO for minimization also require $K$ to satisfy an inequality of the form $K \geq C \kappa^a \log(n^{b} K)$ in order to demonstrate that RR/SO converges enjoys a faster convergence rate (in terms of the dependence on $n$ and $K$) than uniform sampling. [1, 2, 3]
>
> - [1] K. Ahn, C. Yun, S. Sra (2020) "SGD with shuffling: optimal rates without component convexity and large epoch requirements". Neural Information Processing Systems, 2020.
>
> - [2] K. Mishchenko, A. Khaled, P. Richtarik (2020) "Random Reshuffling: Simple Analysis with Vast Improvements". Neural Information Processing Systems, 2020.
>
>
> - [3] D. Nagaraj, P. Jain, P. Netrapalli (2019) "SGD without Replacement: Sharper Rates for General Smooth Convex Functions". International Conference on Machine Learning, 2019.

---

> > ### Comment · Reviewer_KJCr · 2022-08-06
> > **Concerns adressed**
> >
> > I thank the authors for their detailed reply.
> >
> > I did not have a lot of concerns, and they have all been addressed. I will raise my score to 7.
> >
> > However, I would like to point one little thing to authors. While they are right about the fact l.704 is well defined, the differentiability almost everywhere is not a sufficient argument. This is also used in l.649 where they say $\nu$ is almost everywhere differentiable, hence l.649 is well defined. The integral is indeed well defined, but this argument is not sufficient to conclude on the equality above l.649. Indeed, one can think of $\nu(t) = \mathbb{1}_{t \geq 1/2}$, a function that is almost everywhere differentiable with derivative 0 almost everywhere, hence the integral over $[0, 1]$ would not be $\nu(1) - \nu(0)$. The Lipschitz continuity argument makes it work, but must not be used only to conclude on the differentiability almost everywhere.
> >
> > Moreover, $\nu(z^*)$ disappeared in the same equality (above l. 649).

---

> ### Author Response · Authors · 2022-08-02
> **Regarding Other Issues and Suggestions**
>
> # Twice Differentiability Requirement in Line 674 (Line 704 in updated version)
>
> As a consequence of Rademacher's Theorem, the Lipschitz continuity of $\omega_{\tau_k(i)}$ implies that $\omega_{\tau_k(i)}$ is differentiable almost everywhere (with respect to the Lebesgue measure). Thus, by property of Lebesgue integrals, both $M_{\tau_k(i)}$ and $J_{\tau_k(i)}$ are well defined (without any need for assuming twice differentiability of $f_i$). As stated in Lines 705-707 of our (updated) manuscript, this line of reasoning has been elucidated in Theorem C.1.
>
> # Loose Inequalities in Line 719 (Line 760 in updated version)
>
> We agree that the inequalities here can be tightened. However, in our experience, using tighter inequalities for bounding the sum does not improve the dependence of $\alpha$ on $l$, $\mu$, and $n$, and can lead only to constant factor improvements (this is also intuitive since the effective step-size of the epoch level update rule is $n \alpha$ and full batch GDA needs step sizes of the order $\mu/l^2$ for convergence). On the contrary, the use of tighter inequalities leads to a considerably more cumbersome presentation. Hence, for the sake of clarity and accessibility, we opt in favor of looser inequalities for our presentation.
>
> # Remaining Suggestions
>
> Thank you for pointing these out. We have incorporated your suggestions in the updated version of our paper.

---

### Meta-Review · Area_Chair_vgZM · 2022-08-26

**Recommendation:** Accept
**Confidence:** Less certain

**Metareview:**

All reviewers acknowledge that the paper fills a gap in the literature, with good results for a wide variety of settings.

**Award:**

No

---

### Decision · Program_Chairs · 2022-09-14

Accept